# Reconfigurable Stochastic neurons based on tin oxide/MoS$_2$ hetero-memristors for simulated annealing and the Boltzmann machine

Xiaodong Yan [1,6], Jiahui Ma[1,6], Tong Wu [2], Aoyang Zhang[1], Jiangbin Wu [1], Matthew Chin[3], Zhihan Zhang[4], Madan Dubey[3], Wei Wu [1], Mike Shuo-Wei Chen[1], Jing Guo [2] & Han Wang [1,5✉]

Neuromorphic hardware implementation of Boltzmann Machine using a network of stochastic neurons can allow non-deterministic polynomial-time (NP) hard combinatorial optimization problems to be efficiently solved. Efficient implementation of such Boltzmann Machine with simulated annealing desires the statistical parameters of the stochastic neurons to be dynamically tunable, however, there has been limited research on stochastic semiconductor devices with controllable statistical distributions. Here, we demonstrate a reconfigurable tin oxide (SnO$_x$)/molybdenum disulfide (MoS$_2$) heterogeneous memristive device that can realize tunable stochastic dynamics in its output sampling characteristics. The device can sample exponential-class sigmoidal distributions analogous to the Fermi-Dirac distribution of physical systems with quantitatively defined tunable "temperature" effect. A BM composed of these tunable stochastic neuron devices, which can enable simulated annealing with designed "cooling" strategies, is conducted to solve the MAX-SAT, a representative in NP-hard combinatorial optimization problems. Quantitative insights into the effect of different "cooling" strategies on improving the BM optimization process efficiency are also provided.

[1] Ming Hsieh Department of Electrical and Computer Engineering, University of Southern California, Los Angeles, CA 90089, USA. [2] Department of Electrical and Computer Engineering, University of Florida, Gainesville, FL 32611, USA. [3] Sensors and Electron Devices Directorate, U.S. Army Research Laboratory, Adelphi, MD 20723, USA. [4] School of Electrical and Computer Engineering, Georgia Institute of Technology, Atlanta, GA 30332, USA. [5] Mork Family Department of Chemical Engineering and Materials Science, University of Southern California, Los Angeles, CA 90089, USA. [6] These authors contributed equally: Xiaodong Yan, Jiahui Ma. ✉email: han.wang.4@usc.edu

Stochastic neuron devices are essential for the neural network implementation of key emerging non-von-Neumann computing concepts such as the Boltzmann machines, which are recurrent artificial neural networks with stochastic features analogous to the thermodynamics of real-world physical systems. BM can be used to solve a broad range of combinatorial optimization problems[1,2] with applications in classification[3], pattern recognition[4], feature learning, and other emerging computing systems. Deriving its name from the Boltzmann distribution of statistical mechanics, BM possesses an artificial notion of "temperature", and the controlled evolution of this "temperature" parameter during the optimization process[5,6], i.e., the "cooling" strategy, can impact the convergence efficiency of the BM and its chance of reaching a better cost-energy minimization (or maximization depending on problem definition). To realize the hardware implementation of the BM that can also allow the "temperature" control and hence the precise execution of desired "cooling" strategy, it is essential to have electronic devices that can generate exponential-class stochastic sampling with dynamically tunable distribution parameters.

The property of memristor in its deterministic form has been commonly used in applications such as multiply-and-accumulate matrix calculation[7] and resistor-logic demultiplexers[8–10]. Its stochastic property is often intentionally suppressed[11–13] in such applications for the purpose of achieving accurate and reproducible computational results[14,15]. On the other hand, rich stochastic property of memristors, which relies on ensembles of random movements of atoms and ions, offers opportunities in energy-efficient computing applications[16–20]. With the stochastic property, one can generate random number[21] to encrypt information, implement physical unclonable functions[22], and realize artificial neurons[23] with integrate-and-fire activations. Furthermore, emerging computing schemes can use stochastic memristive device as a building block to emulate biological neural network[24,25], whose

functions—such as decision-making—can leverage the stochastic dynamics of neurons and synapses. However, a common challenge with previous stochastic memristors is the lack of means to precisely control and modulate the probability distribution that is associated with its randomness. Realizing such devices has been difficult because many device-generated random features in stochastic memristors or oscillators lack stable probability distribution, which limits the chance of controlling it experimentally[19,26,27]. Additionally, with only two terminals in a common memristor, where the probability distribution can only be influenced through the two-terminal bias, the probability distribution of the device output cannot be tuned flexibly and precisely.

In this work, we overcome such challenge with a three-terminal stochastic hetero-memristor based on tin oxide/MoS$_2$ heterostructure, which demonstrates tunable statistical distributions enabled by the gate modulation. The inherent exponential-class stochastic characteristics of the device arising from the intrinsic randomness and energy distribution in its ionic motions are explored to realize sampling of exponential-class sigmoidal distributions that resembles the Fermi–Dirac distribution in physical systems. The device incorporates gate modulation that allows the efficient control of the stochastic features in the device output characteristics. The device enables the realization of reconfigurable stochastic neuron and the implementation of Boltzmann machine in which the reconfigurable statistic of the device allows different "cooling" strategies to be implemented during the optimization process. The effect of different "cooling" strategies on improving the optimization process efficiency of the BM is demonstrated experimentally.

## Results

Figure 1a shows the schematic of this reconfigurable hetero-memristor, where tin oxide serves as filament-switching layer and

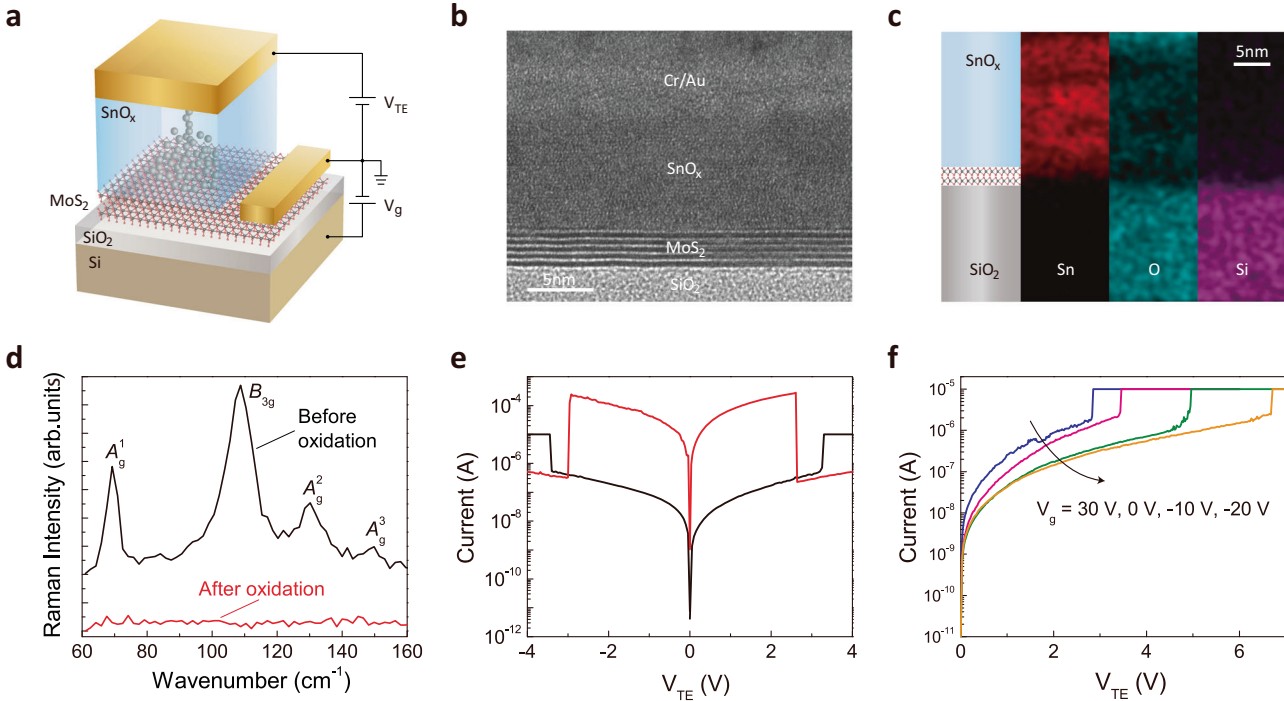

**Fig. 1 Device structure and electrical characteristics. a** Schematic of the heteromemristive device. **b** The HR-STEM image of the fabricated device cross section. The scale bar is 5 nm. **c** EDX scan indicates the elemental composition. **d** Raman spectra for the SnSe sample before and after oxidation. The missing-signature modes after oxidation indicate the full oxidation and amorphization of the SnSe sample. **e** Unipolar electrical switching characteristics of the device at $V_g = 0$ V. The set and reset voltages in positive scan are 3.2 V and 2.8 V, and in negative scan are −3.4 V and −3 V. **f** Modulation of the set voltage by the gate bias. When $V_g$ decreases from 30 V to −20 V, the set voltage increases.

is sandwiched between a $MoS_2$ layer and Cr/Au top electrodes (TE). The Si substrate serves as a modulating gate bias ($V_g$) that can influence the filament-formation dynamics in the tin oxide layer. The high-resolution scanning transmission electron microscopy (HR-STEM) image in Fig. 1b shows the cross section of the fabricated device and reveals that the tin oxide layer is amorphous. An energy-dispersive X-ray spectroscopy (EDX) scan in Fig. 1c indicates the elemental composition. Figure 1d plots the Raman spectra for the SnSe sample before and after oxidation, which leads to the formation of the $SnO_x$ layer. All signature modes of SnSe, including the shear mode $A_g^1$, the in-the-plane modes $A_g^2$ and $B_{3g}$, and the out-of-plane mode $A_g^3$ that are observed before oxidation, and are not detected after oxidation, indicating the full oxidation and amorphization of the SnSe sample[28]. The tin oxide film can also be synthesized using atomic-layer deposition (ALD)[29–31], which produces films of similar quality as the direct oxidation method.

Unipolar electrical switching characteristics of the device at $V_g = 0$ V are shown in Fig. 1e. It sets and resets at around 3.2 V and 2.8 V respectively in the positive bias, and at −3.4 V and −3 V, respectively, in the negative bias[32]. Both the Joule heating and the electric-field driven effect can be playing roles in the device operation. The filament-formation operation can be due to a breakdown-like process with random creation of voltage-stress-induced vacancy or defect sites, which is electric-field driven. The Joule heating can be the main effect in filament rupturing. The insertion of the $MoS_2$ layer in the device made it possible to adjust the electron energy level in $MoS_2$ by externally modulating the gate bias $V_g$, which can modulate both the contact-energy barrier between the $MoS_2$ and $SnO_x$, and the conductivity of the $MoS_2$ sheet itself (see supplementary information section 4). Hence, as shown in Fig. 1f, as the gate bias decreases from 30 V to −20 V, the electrostatic doping in $MoS_2$ and the associated energy level decreases, leading to the reduction in the series conductivity and hence the gradual increase in the set voltage.

The filament-formation process is stochastic due to the inherent random motion of oxygen ions. To extract this stochastic property quantitatively, a statistical study is carried out on the set process. As shown in Fig. 2, the device is initially reset to the high-resistance state and a bias $V_{TE}$ is applied to the device for up to 2 s. During each set process, it takes a certain amount of time $t$ ($t \leq 2$ s) after the bias voltage is applied for the device to be set. This required bias time until set is stochastic in each trial. Furthermore, there is certain chance that the device may still remain in the high-resistance state after 2 s. Figure 2a plots the device current characteristics as a function of time when this reset and set process was repeated for 30 times at $V_{TE} = 6$ V, 5 V, 4 V, and 3 V, respectively, with $V_g$ fixed at 0 V. At $V_{TE} = 6$ V, the device is successfully set within the first 2 s for all the 30 trials. At $V_{TE} = 5$ V, 4 V, and 3 V, the device failed to set within the first 2 s in certain cases. Figure 2b shows the histogram probability distribution extracted from 30 trials of the time required, until the device becomes set. If we consider $t$ as a random variable, the probability that the set will occur within an infinitesimal interval $\triangle t$ at time $t$ can be described by an exponential-class distribution[33] function $P = \frac{\triangle t}{\tau} \cdot e^{-\frac{t}{\tau}}$ with the wait time $t$ following a Poisson distribution (see supplementary information section 6) and it fits the experimental data well (red lines, Fig. 2b). This experimental observation resembling Poisson random wait time underlying the filament-formation process in the tin oxide memristive device is indicative of its exponential-class stochastic nature.

Moreover, Fig. 2c plots $P_{ss,t<2s}$ as a function of $V_{TE} - V_{TE0}$ under different gate voltages, which shows exponential-class sigmoidal distribution function. Here, $P_{ss,t<2s}$ is the probability

that the device will successfully set within 2 s and $V_{TE0}$ is the 50% probability bias-voltage point, i.e., $P_{ss,t<2s}$ ($V_{TE} = V_{TE0}) = 0.5$. With the gate voltage fixed, the chance of the device being set within $t < 2$ s becomes higher with increasing $V_{TE}$, following a sigmoidal distribution. It shows that $V_{TE}$ can tune the stochastic property of the set event in the device when $V_g$ is fixed. Microscopically, the $V_{TE}$ tunes the filament-formation process by modulating the vacancy-hopping barrier height and thus the ion-hopping rate. Thus, the device is understandably easier to set at high $V_{TE}$ than low $V_{TE}$. Under different gate voltages, $P_{ss,t<2s}$ shows a sharper 0-to-1 transition when $V_g$ is 30 V and a wider spread in its 0-to-1 transition when the $V_g$ decreases. Here $V_g$ tunes the Fermi level and charge density in the $MoS_2$ layer, which modulates the potential distribution between $MoS_2$ and tin oxide layer under $V_{TE}$ bias. $V_{TE}$ is more effective in modulating the device when $V_g$ is higher, i.e. the $MoS_2$ layer has a higher electron carrier density and higher conductivity, and thus leads to a sharper 0-to-1 transition in the sigmoidal distribution curve.

The set process is achieved by the filament formation through stochastic vacancy generation and hopping-transport processes. Applying a voltage can reduce the generation and hopping-barrier height and exponentially enhance the generation and hopping rates. Analytically, the set probability, $P_{ss,t<2s}$, can be derived as $P_{ss,t<2s} = 1 - e^{-\beta e^{\alpha(V_{TE} - V_{TE0})}}$, where $\alpha$ and $\beta$ are parameters related to the material and device structure (see supplementary information section 7). After further approximation, $P_{ss,t<2s}$ can be simplified to a distribution function that resembles the Fermi–Dirac distribution (see supplementary information section 8):

$$P_{ss,t<2s} \approx \frac{1}{1 + \exp\left(-\frac{V_{TE} - V_{TE0}}{T_{eff}}\right)} \quad (1)$$

where $T_{eff}$ is an effective "temperature" term that can be tuned by the gate bias. This expression fits very well with the experimental data in Fig. 2c. The above analytical description is also in agreement with kinetic Monte Carlo simulations, which describes microscopic stochastic process of vacancy generation, hopping, and recombination in filament formation[34,35]. $T_{eff}$ corresponding to various gate voltages is extracted from the fitting and Fig. 2d plots $T_{eff}$ versus gate voltage $V_g$. A behavioral model is developed to understand the dependence of the $T_{eff}$ on the gate-bias voltage. The device is modeled as a memristor in serial combination with a $MoS_2$ layer whose resistance (both the sheet resistance and its contact property with the memristive filament) can be modulated by the gate electric field. As a result, $T_{eff}$ can be expressed as $T_{eff}\left(V_g\right) = T_{V0}\left[1 + \frac{Z}{(V_g - V_T)}\right]$, where $T_{V0}$ and $Z$ are constants, $V_T$ is the threshold voltage (see supplementary information section 9). As shown in Fig. 2d, this model fits well with the experimental data and describes the modulation effect of $T_{eff}$ by $V_g$. We would like to note that the value of $T_{eff}$ has the unit of volt. However, to avoid confusion with the actual electrical bias voltages applied on the device, the unit of $T_{eff}$ will be omitted in the subsequent discussions. The above discussed stochastic process of the filament formation together with the gate voltage-dependent "temperature" effect can be used to construct exponential-class distribution sampling that has broad applications in statistical modeling and computing, with the Boltzmann machine as a typical example.

To demonstrate the unique advantages of these tunable exponential-class stochastic heteromemristors in computing application, a version of Boltzmann machine that contains a network of stochastic neurons is implemented. The stochastic neurons may fire in response to the input signals and thus drive the searching dynamics of the BM. The BM iterates all possible solutions to search for the best solution by minimizing the

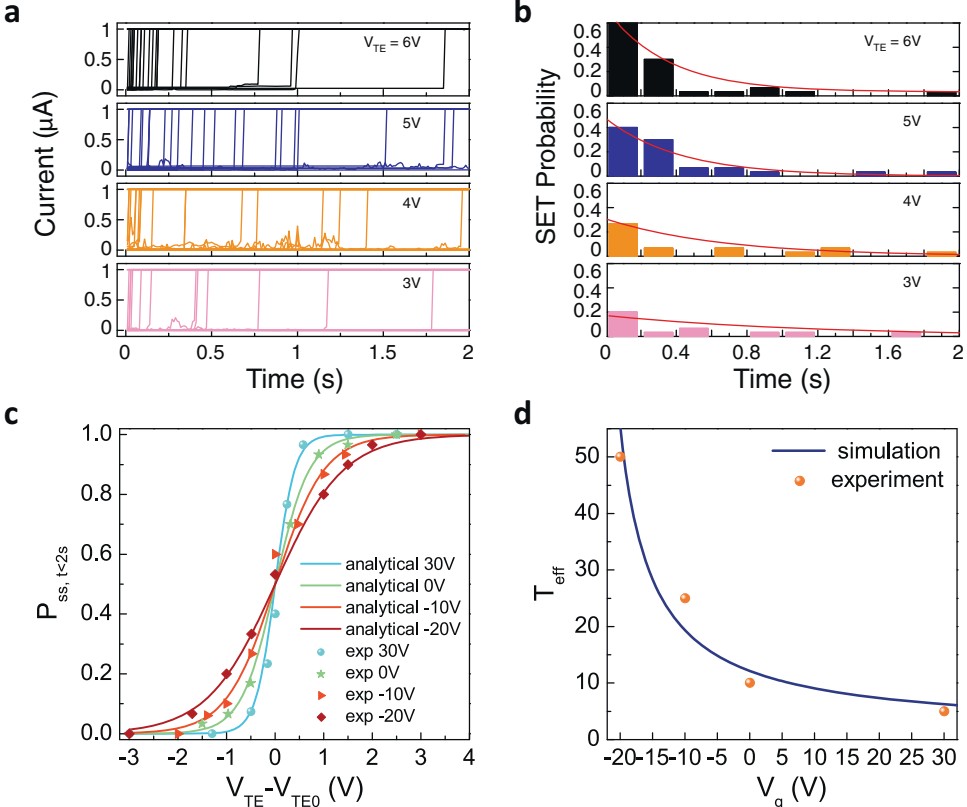

**Fig. 2 Sampling of exponential-class sigmoidal distribution. a** The set process under different $V_{TE}$. The initial state is reset to high-resistance state and a bias $V_{TE}$ is applied to the device for 2 s. **b** The experimentally extracted probability distribution of the bias time until set occurrence for $V_{TE} = 3$ V, 4 V, 5 V, and 6 V, respectively. **c** $P_{ss,t<2s}$ as a function of the $V_{TE}$ under different gate voltages, showing exponential-class sigmoidal distribution function. Experimental results are shown as data symbols, and the analytical model fit is shown in lines. **d** Experimental results (dots) and model fit (line) showing the relation between $T_{eff}$ and the gate bias $V_g$.

system-energy function. Hardware implementations[36,37] of such BM are challenging with conventional transistors and would require a large number of devices and complex circuitry. Here we build a BM where each of the stochastic neuron is based on a single tin oxide/MoS$_2$ hetero-memristor as stochastic switching and simple peripheral circuitry (more details in Methods: BM construction). This implemented BM is used to solve a maximum satisfiability problem (MAX-SAT), which is an NP-hard combinatorial optimization problem underlying a wide range of key applications, including Max-Clique[38], correlation clustering[39], treewidth computation[40], Bayesian network structure learning[41], and argumentation dynamics[42].

Given a set of Boolean clauses, where each clause is a disjunction of Boolean variables and their negations, the MAX-SAT problem[43] aims to maximize the number of clauses that can be true when truth values are assigned to the Boolean variables. Without the loss of generality, the set of Boolean clauses to be solved in this work are selected to be {Ci|i = 1, 2, ... , 5}, where the clause C1 is $(x \lor y \lor z)$; C2 is $(x' \lor y \lor z)$; C3 is $(x' \lor y' \lor z)$; C4 is $(x \lor y' \lor z')$ and C5 is $(x' \lor y \lor z')$ (shown in Fig. 3a, the Boolean variable $x'$ is the negation of the Boolean variable $x$). The optimization task here is to find a state vector $\mathbf{X} = (x_1, \cdots, x_6) = (x, y, z, x', y', z')$ that can maximize the number of clauses to be true. A MAX-SAT can be converted equivalently to a problem that is solvable for the BM[44,45]. Six stochastic units are used in the BM to realize the activation for each Boolean variable in the state vector $\mathbf{X} = (x_1, \cdots, x_6)$. Then we build a weight matrix $\mathbf{W}$. The weight $w_{ij}$ that is between every

two Boolean variables is assigned based on the MAX-SAT problem. Solving the MAX-SAT is equivalent to minimizing the total energy $E = \mathbf{X}^T \mathbf{W} \mathbf{X}$ of the BM, where $\mathbf{X}^T$ is the transverse of $\mathbf{X}$.

The constructed BM utilizing the tin oxide/MoS$_2$ hetero-memristors is shown in Fig. 3b and the schematic of the circuit blocks with six stochastic neurons is shown in Fig. 3c. In each iteration step, if the hetero-memristor sets, the Boolean value of $x_i$ would be flipped. If the heteromemristor does not set, the stochastic neuron would not fire and $x_i$ remains the same. The stochastic neurons are sequentially updated until the BM reaches the optimal solution. In Fig. 3d, we experimentally demonstrated the evolution of the state vector and total energy when the BM started from three different initial states and found the same optimal solution, which is $\mathbf{X} = (x, y, z, x', y', z') = (0, 1, 1, 1, 0, 0)$.

As previously shown in Fig. 2d, $V_g$ can tune the tin oxide/MoS$_2$ heteromemristor to have different $T_{eff}$ during the BM optimization process. $T_{eff}$ of the BM describes the average behaviors of all the stochastic units, in close analogy to the temperature parameter in the Boltzmann distribution that describes the average behavior of particles under different thermal equilibrium states in physical systems. Thus, by controlling $T_{eff}$ in the optimization process that can be achieved via tuning the $V_g$, it is possible to avoid premature convergence issues and facilitate the convergence efficiency associated with the BM. Figure 3e shows the effect of different $V_g$ bias on the BM optimization process. During these three different runs of the BM, all the tin oxide/MoS$_2$ stochastic hetero-memristors are biased at $V_g = -20$ V, 0 V, and 20 V, respectively. The energy evolved differently during these

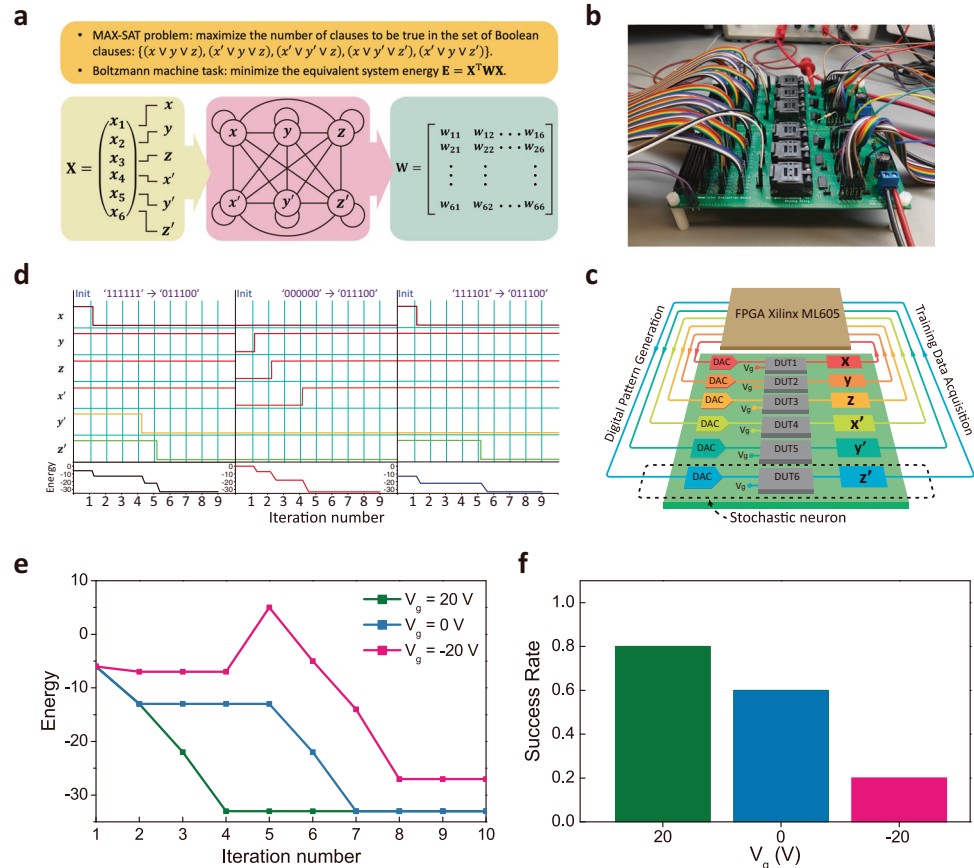

**Fig. 3 Boltzmann machine implementation using tin oxide/MoS₂ heteromemristor. a** Flow chart showing the steps in mapping a MAX-SAT problem to an equivalent form solvable using the Boltzmann machine. **b** The PCB evaluation board of BM-integrated system, including the packaged tin oxide/MoS₂ memristive units and CMOS peripheral circuits. **c** Schematic of the BM circuit blocks with six tin oxide/MoS₂ heteromemristors as the artificial neurons. **d** The experimentally obtained evolution of state vector and total energy when the BM was started from three different initial states, resulting in the same optimal solution. **e** Experimentally obtained energy evolution in the BM optimization process with $V_g = -20$ V, 0 V, and 20 V, respectively. **f** The success rate of the BM optimization process under different $V_g$.

runs each time. The BM is at $T_{eff} = 7$ when $V_g = 20$ V and converges easily for this particular problem. On the other hand, the BM is at $T_{eff} = 50$ when $V_g = -20$ V and is less efficient in reaching convergence. For $V_g = 0$ V, the BM is at $T_{eff} = 10$ and converges at an intermediate rate among the three cases. By counting how many times the BM can reach the global optimal solution out of 50 trial runs, the success rate as a function of $V_g$ and $T_{eff}$ is statistically obtained as shown in Fig. 3f. It indicates that the $V_g$ and hence the $T_{eff}$ can substantially affect the performance of the BM.

Simulated annealing[46,47] can be implemented with our BM where the $T_{eff}$ can gradually change during the optimization process to emulate different "cooling" strategy. It is an important approach for efficiently reaching better optimization solutions and for avoiding the premature convergence. Using the gate-tunable tin oxide/MoS₂ device, such "cooling" procedures can be quantitatively implemented during the simulated annealing by translating the designated sequential evolution of $T_{eff}$ into the corresponding series of gate voltage bias conditions following the relation in Fig. 2d. To study the effect of different "cooling" strategies on the efficiency of the BM, four different $T_{eff}$ variation strategies were experimentally applied on the BM. Strategy 1: high $T_{eff}$ in the first three iteration steps followed by low $T_{eff}$ for the remaining iterations in one optimization process (HT to LT), Strategy 2: low $T_{eff}$ in the first three iterations followed by high $T_{eff}$ for the remaining iterations (LT to HT), Strategy 3:

maintaining a low $T_{eff}$ in the entire optimization process (LT), and Strategy 4: maintaining a high $T_{eff}$ in the entire optimization process (HT). Figure 4a shows the qualitative schematic about how system energy (color dots) would evolve in the process of searching optimal solutions among multiple possible energy minimums (gray line). To analyze the effect of these "cooling" strategies, typical evolutions of the energy (cost function) during the BM optimization process for the four different strategies were experimentally obtained. As shown in Fig. 4b, using the HT strategy ($T_{eff} = 50$), the BM is highly active but loses the selectivity for reaching proper convergence. Using the LT strategy ($T_{eff} = 5$), the BM is significantly less active but possesses higher selectivity that facilitates its convergence to a premature state. Finally, simulated annealing using a "cooling" strategy (HT to LT) enables active initial searches at HT ($T_{eff} = 50$) and then steady convergence to the minimum energy state at LT ($T_{eff} = 5$) as shown in the experimental results. Furthermore, Figs. 4c and 4d show the experimentally obtained statistics of success rate in finding the global optimal solution when the different "cooling" strategies are used. Different initial values for the state vectors are used in Figs. 4c and 4d to show the effect from the different initial conditions. Both figures indicate that the HT to LT strategy has the highest success rate for reaching the global optimal solution for this particular problem, while the HT strategy has the lowest success rate. The results are consistent with the simulated performance of the BM (see supplementary information section 10).

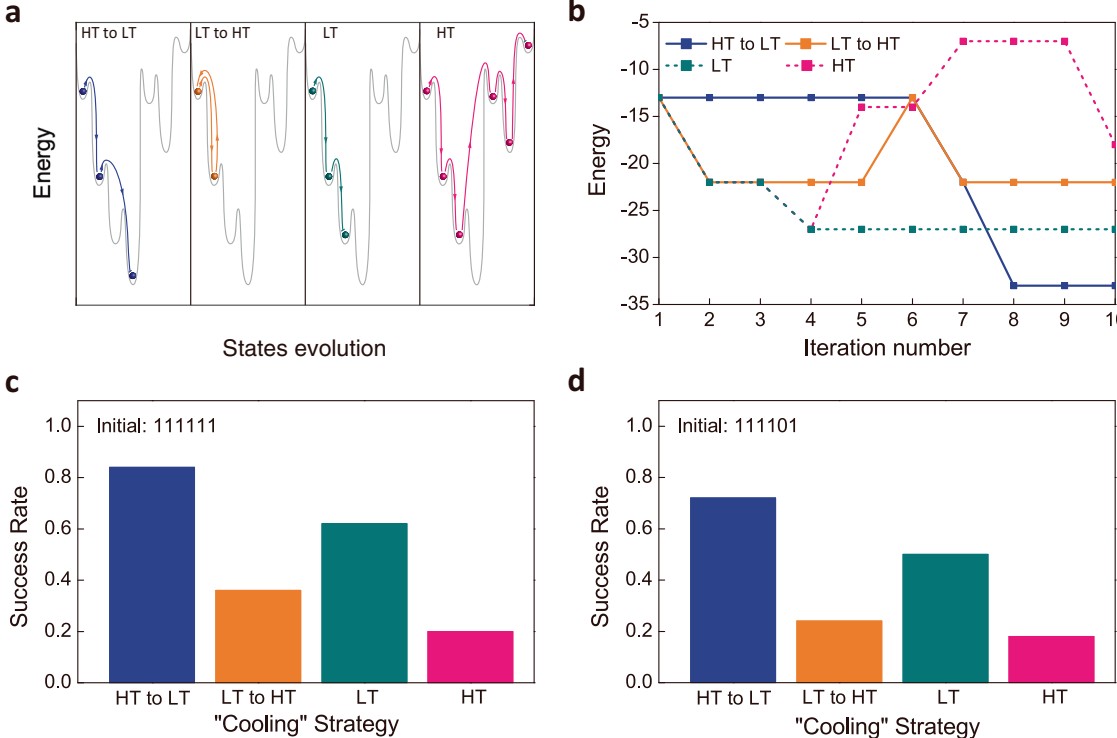

**Fig. 4 Implementing the simulated annealing in tin oxide/MoS$_2$-based BM. a** Conceptual schematic illustrating the evolution of the solution and energy states during the optimization process employing the four different variation strategies. **b** Experimentally obtained energy evolution in the BM optimization process for the four different strategies. **c**, **d** Experimentally extracted success rate of the BM in achieving the most optimal solution using four different strategies of T$_{eff}$ variation during the optimization process: HT to LT, LT to HT, LT, and HT. Different initial states are used in **c** and **d**. T$_{eff}$ = 50 for HT and T$_{eff}$ = 5 for LT in **b**, **c** and **d**.

To quantitatively understand why $T_{eff}$ can make such a significant difference in the BM optimization process, we analyze the Russel–Rao (RR) similarity[48] between all the clauses for this particular MAX-SAT problem. It is because, as illustrated in Fig. 5a, all the five clauses C1–C5 bear inherent similarity to each other due to the following two constraints: the variable constraint and the clause constraint. On the variable side, a Boolean variable and its negation (two variables connected by red lines) are always logically opposite. For example, $x$ and $x'$ will always have opposite values. On the clause side, the chance of two clauses both being true is lower if they contain more complementary Boolean variables in each clause. By assigning true values to the variables $x$, $y'$ and $z'$(yellow circle), the number of complementary variables (blue circle) between clauses could be easily observed. Counting the number of complementary variables can directly reflect the inner connection and constraint of the clauses. In Fig. 5a, for example, if the clause C4: $(x \lor y' \lor z')$ is true, then the probability that the clause C2: $(x' \lor y \lor z)$ also being true is much smaller than the other three clauses since C4 and C2 contain three pairs of complementary variables.

With the BM set to different $T_{eff}$, the RR similarity matrix among the five clauses based on the experimental data is constructed in Figs. 5b, 5c and 5d. The color and number in each cell quantify the similarity between each pair of clauses indexed by the row and column. It represents the probability when both clauses are true among all cases. For example, a RR similarity of 0.84 between C1 and C2 in Fig. 5b means that by repeatedly running the BM 50 times at $T_{eff} = 50$, we had C1 and C2, both being true by the end of 42 (out of 50) runs.

The effect of $T_{eff}$ can be explained as follows. We view the RR similarity as the distance measurement of the statistical relationship between each of the two clauses (distance = 1 − RR

coefficient) in solution space[49]. In other words, clauses with RR similarity close to 1 are seen as closely clustered, while the clauses with RR similarity close to 0 are furthermost separated. When $T_{eff}$ is tuned to 50 (Fig. 5b), all the clauses have similar distances in the solution space, since they show close RR similarity between all pairs. As a consequence, BM tends to search widely in the solution space with a high robustness, high stochasticity, and low selectivity, since choosing any solution would look the same to the BM. When $T_{eff}$ is 20 (Fig. 5c), clauses with small distances are closely clustered, giving high RR similarity close to unity for pairs of clauses that can be easily satisfied simultaneously, such as C1 and C2, and a low RR similarity for pairs of clauses that can hardly be satisfied at the same time, such as C1 and C4. At this $T_{eff} = 20$, the BM gains more selectivity in solution space. When the $T_{eff}$ is 5 (Fig. 5d), all the clauses are either strongly clustered or separated in distance, with distinct either 1 or 0 RR similarity. BM behaves more like a deterministic "machine". This tends to cause premature convergence as the BM is significantly less active.

Next, a simulated annealing process in the BM with linear cooling is simulated in Fig. 5e. The evolution of the RR similarity matrix indicates that the BM would evolve through all the cases that are discussed above from being fully stochastic toward nearly deterministic as $T_{eff}$ decreases linearly. Thus, the simulated annealing process of a BM could be understood as such: at high $T_{eff}$, the BM searches solution space globally with high robustness and low selectivity, for the sake of large gradient descent; as the BM cools down, it gains selectivity toward some solutions and can possibly jump out of local minima since $T_{eff}$ still provides enough perturbation; as the BM cools down to the limit, the BM exhibits a stronger selectivity than robustness, preventing itself from jumping out of the optimal zone. Hence, more efficient

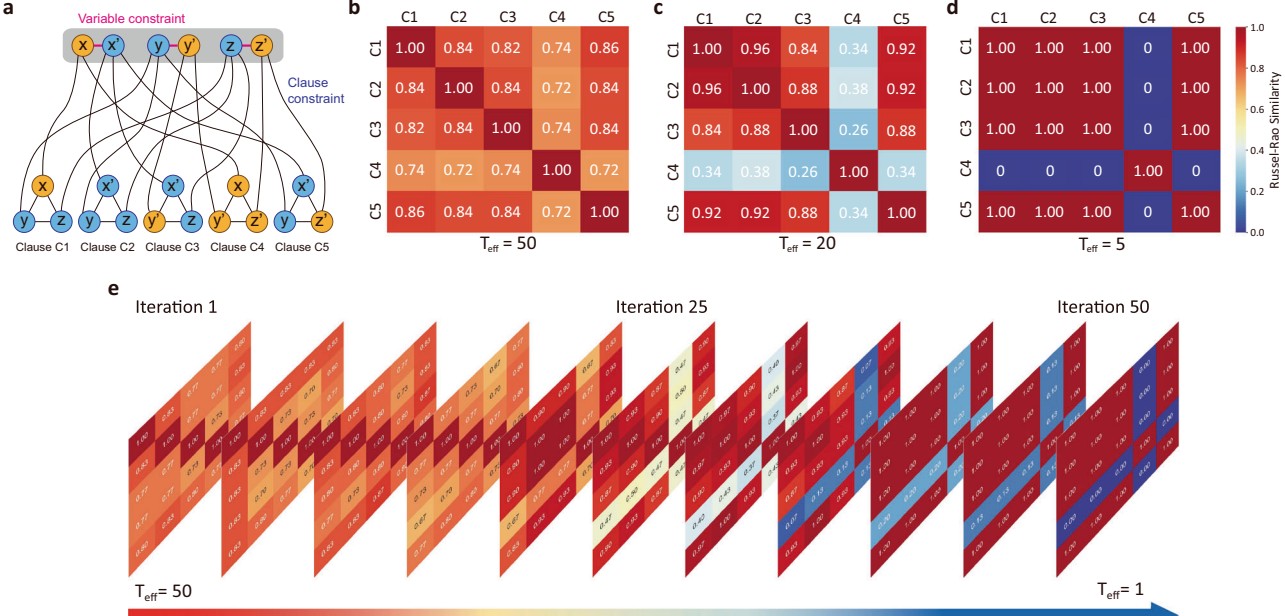

**Fig. 5 Russel–Rao similarity matrix underlying the clauses employing different "cooling" strategy in a MAX-SAT problem. a** Schematic shows that the five clauses in MAX-SAT problem are correlated with each other, which is imposed by variable constraint and clause constraint. For illustration, the yellow-circled variables are assigned true values, thus making the blue-circled variables false. **b c d** Russel–Rao similarity matrix between the five clauses when BM runs the optimization process under $T_{eff} = 50$, 20, and 5, respectively. **e** The evolution of the Russel–Rao similarity matrix in a BM optimization process when $T_{eff}$ is decreased linearly with each iteration step.

performance in the BM can be achieved with an appropriate "cooling" strategy.

In summary, tunable stochastic behavior is demonstrated in the tin oxide/$MoS_2$ heteromemristor, showing inherent exponential-class statistical characteristics. The device can sample exponential-class sigmoidal distributions resembling the Fermi–Dirac distribution in physical systems with tunable distribution parameters to emulate the "temperature" effects. Simulated annealing with control of the "cooling" strategies is demonstrated in the implemented Boltzmann machine for solving combinatorial optimization with respect to a MAX-SAT problem. These stochastic neurons based on tin oxide/$MoS_2$ heteromemristors with reconfigurable statistical behavior pave the way for implementing selected "cooling" strategies in BM to reach optimal convergence efficiency and can find broad applications in energy-efficient computing for learning, clustering, and classification.

## Methods

**Device fabrication**. A thin $MoS_2$ layer is first deposited on a Si wafer with a 285-nm thermally grown $SiO_2$ layer on top. The sample is then treated in an $Ar/H_2$-mixed gas environment at 350 °C to clean the $MoS_2$ surface. Subsequently, a thin tin oxide oxidized from SnSe is deposited on $MoS_2$ and serves as filament-switching layer. Electron beam lithography is then used to transfer the patterns followed by the evaporation of a 10-nm/40-nm Cr/Au metal stack, which forms the top electrode.

**STEM and EDX**. A FEI Titan Themis G2 system was used to prepare the HRSTEM images with four detectors and spherical aberration. To observe the cross-section image, the sample was pretreated by depositing chromium and carbon-capping layers, then thinned by a focused-ion beam (FIB, FEI Helios 450 S) with an acceleration voltage of 30 kV. The HRSTEM image was acquired with an acceleration voltage of 200 kV. EDX signals were collected to identify the elemental component in the cross section, which was integrated within the STEM system.

**Raman spectroscopy**. A Renishaw inVia Qontor system was used to measure the Raman spectra, which was installed with a ×100 objective lens, a grating (1800 grooves $mm^{-1}$), and a charge-coupled device camera. The wavelength of the

excitation laser was 532 nm (from a solid laser). The Raman spectra resolution is 1.2 $cm^{-1}$ per pixel.

**BM construction**. The implemented BM prototype contains 24 5-bit digital-to-analog converters (DAC). The digital pattern generation interface (DPGI) and training data acquisition interface (TDAI) are controlled by a Xilinx ML605 FPGA board that carries out information storage and computations. It formed a feedback loop to adjust both input and output patterns at each BM iteration. Depending on different input signals, the BM system adjusts the corresponding output training data accordingly. The BM prototype has six stochastic units, with each unit containing a tin oxide/$MoS_2$ heteromemristor that has approximately sigmoidal switching probability upon applied voltages and peripheral circuitry. The peripheral circuitry is consisting of 4 DACs (digital-to-analog converter) to read digital voltage values and apply to heteromemristor, a dynamic comparator for generating discrete-state readout and output-level shifters.

## Data availability

The data that support the findings of this study are available from the corresponding author upon reasonable request.

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

## Acknowledgements

This work is supported in part by the Army Research Office (grant no. W911NF-21-2-0128) and National Science Foundation (grant no. CMMI-2036359). T.W. and J.G. acknowledge support by National Science Foundation (grant no. 1809770 and 1904580). W.W. acknowledges the support from Air Force Research Laboratory (grant no. FA8750-19-1-0503).

## Author contributions

X.Y., J.M., and H.W. conceived the project idea. X.Y., J.M. and J.W. fabricated the devices, characterized their electrical performance, and constructed and measured the BM circuit. A.Z., X.Y., M.S.-W.C., and Z.Z. contributed to the design of the BM circuit. M.C and M.D. contributed to the device fabrication. W.W. contributed to the understanding of the device operation. T.W, X.Y., J.M., and J.G led the simulation and modeling of the device and BM circuit. H.W. coordinated and supervised the overall research activities. All coauthors contributed to the discussion of the data. X.Y., J.M., T.W., J.G., and H.W. cowrote the paper with inputs from all coauthors.

## Competing interests

The authors declare the following competing interests: H.W. currently also leads the low-dimensional materials research at Taiwan Semiconductor Manufacturing Company (TSMC) Corporate Research. All other authors declare no competing interests.
