## [Peer Review File · Nature Communications]

Reviewers' Comments:

Reviewer #1:

Remarks to the Author:

This manuscript by Yan et al reported a three-terminal tin oxide (SnO_x)/molybdenum disulfide (MoS_2) heterogeneous memristive device with the tunable stochasticity and its based the Boltzmann machine (BM) application. The idea of using the three-terminal memristor as tunable stochastic neuron for the BM seems interesting and novel, and the BM integrated system have been demonstrated based on the PCB evaluation board with the fabricated devices. However, although one of the major novelties in this manuscript is to demonstrate the tunable stochastic memristor-based BM, many parts of the memristor do not meet an objective standard. For example, the authors insisted that the unipolar switching behavior of the fabricated device is due to "the field-assisted drift of oxygen ions" and its based the growth and rupture of the oxygen-vacancy filament; however, the unipolar switching behaviors is generally attributed to the Joule heating, rather than the field-assisted drift that induce the bipolar switching in most cases. And, there is no direct observation of this conductive filament driven by the oxygen vacancies It needs to show the cross-sectional TEM image or evidence of the conductive filament for understanding the switching mechanism since this junction structure is different as compared to other reports.

In addition to this, they just mentioned that both the conductivity of the MoS_2 sheet itself and the interfacial barrier at $\text{MoS}_2/\text{SnO}_x$ can be modulated by V_g , even without any energy band diagrams and references. And, there is no optical/SEM image for top-view of the device. How can make the filament inside the SnO_x when the voltage is applied to semiconducting (non-conducting) MoS_2 and top metal electrode? The probing ways and forming process are missing.

This reviewer suggest that the authors must supplement the switching mechanism and the origin of the gate-tunable characteristics by including various experiments and discussions. Taken all together, the current manuscript is required to be significantly improved overall before it is published in Nature Communications. There are additional comments and suggestions that might help to improve the manuscript as below:

(1) The detailed probing methods and measurements in Fig. 1a should be clearly provided (e.g., an optical image). Also, the authors should identify electrical characteristics of the SnO_x memristor without the insertion of MoS_2 by applying V_g .

(2) In the sampling of exponential-class sigmoidal distribution, a statistical study was performed by switching from the high resistance state to the low resistance state for up to 2 second. This seem to be very slow. The authors should address the slow switching speed and discuss the effect of the slow sampling characteristics on the BM operation.

(3) $V_g = -20 \text{ V} - 30 \text{ V}$ is very high, which can lead to the high power dissipation by V_g due to gate leakage current and gate charging process. Any comment about this issue?

(4) This reviewer did not find the endurance parameter in this manuscript. Because the multiple

SET/RESET operation is required for the flip of Boolean value, the endurance seems to be important parameter. Any comment about this?

(5) Can the authors comment any advantage and benefit of the exponential-class sigmoidal for the BM applications?

(6) The authors described the BM operation as a function of run or iteration number. How long the one run or iteration is taken? Because of the slow switching process of the fabricated device, the BM system can be further slow. The author should address this comment.

(7) The baseline obtained from the theoretical calculation or simulation need to be provided in the BM result to compare with the experimental results.

(8) The authors insisted that an appropriate cooling strategy can lead to more efficient performance in the BM. Can the author suggest any guideline for the appropriate cooling strategy?

Reviewer #2:

Remarks to the Author:

This work reports simulated annealing and Boltzmann machine based on three-terminal hetero-memristors. The work are novel and will be of interest to others in the field. It can be published after addressing a few comments below:

- 1, In the device structure, the authors used a thin SnSe oxidized SnOx as the filament layer. How does the oxidation process affect the underlying MoS2? Will other oxide such as HfO2 function similarly?
- 2, Fig.2d, the Teff approaches 50% when Vg=-20V, shall we expect the rapid increase when Vg goes more negative as the simulation shows? Is there any limit of the Teff?
- 3, Fig.3, does it require each DUT to be exactly the same? Or not, will the randomness of DUT itself affect the machine implementation?
- 4, How does this approach compare with other methods using pure RRAM, MRAM or CMOS? What are the main merits?

Point-by-point Response to Reviewers' Comments for manuscript NCOMMS-21-01017

We are grateful to the reviewers and editors for their time and very helpful comments. In response, we address the comments by the reviewers in detail. We always quote the reviewers' comments first.

~~~~~

### Reviewer #1

#### Comments 1:

This manuscript by Yan et al reported a three-terminal tin oxide ( $\text{SnO}_x$ )/molybdenum disulfide ( $\text{MoS}_2$ ) heterogeneous memristive device with the tunable stochasticity and its based the Boltzmann machine (BM) application. The idea of using the three-terminal memristor as tunable stochastic neuron for the BM seems interesting and novel, and the BM integrated system have been demonstrated based on the PCB evaluation board with the fabricated devices.

**Response:** The authors would like to thank the reviewer for the positive and constructive feedback. Based on the reviewer's comments, we have revised the manuscript to further clarify the switching mechanism and the origin of the gate-tunable characteristics. Below, we address the reviewer's comments point by point.

#### Comments 2:

However, although one of the major novelties in this manuscript is to demonstrate the tunable stochastic memristor-based BM, many parts of the memristor do not meet an objective standard. For example, the authors insisted that the unipolar switching behavior of the fabricated device is due to "the field-assisted drift of oxygen ions" and its based the growth and rupture of the oxygen-vacancy filament; however, the unipolar switching behaviors is generally attributed to the Joule heating, rather than the field-assisted drift that induce the bipolar switching in most cases. And, there is no direct observation of this conductive filament driven by the oxygen vacancies It needs to show the cross-sectional TEM image or evidence of the conductive filament for understanding the switching mechanism since this junction structure is different as compared to other reports.

**Response:** We thank the reviewer for the comments. We agree with the reviewer that the field-assisted drift of oxygen ions is not the only mechanism in the device operation and joule heating can also play an important role in the device operation.

On one hand, the device can be reset when the magnitude of the current is high regardless of the current direction, which, as the reviewer has correctly pointed out, indicates that joule heating may play an important role in rupturing of the oxygen-vacancy filament. On the other hand, we believe the field-assisted drift also plays a role in this device, especially in the SET process. For example, during transition from HRS to LRS, the initial current (at HRS) is expected to be very low and the joule heating effect alone would not likely to be sufficient to (re-)form the filament. The electric field within the  $\text{SnO}_x$  layer applied by the bias voltage needs to become strong enough to cause oxygen ions to drift and enables the electromigration of the defects to form the oxygen-vacancy type filament. Previous work in the literature on metal-oxide type memristive devices1 (such as Appl. Phys. Lett. 92, 022110 (2008)) mentioned joule heating can be the main effect in filament rupturing, while the filament formation typically requires electromigration of the defects that is at least partially due to electric field driven effect. In particular, it was pointed out in Ref.[1]

that the extended defects in the oxide material, e.g., dislocations and grain boundaries, can facilitate filament formation and lower the required electric field. Refs.[2,3] also mentioned that in unipolar device, the filament formation operation is still due to a breakdown-like process with random creation of voltage-stress-induced vacancy or defect sites, which is electric-field driven. Hence, we believe both the joule heating and the electric-field driven effect can be playing roles in the device operation.

There are several experimental studies on SnOx based memristive devices4-6, which have demonstrated the existence of oxygen-vacancy filament in SnOx layer. Considering that only a few one-dimensional filaments will be formed inside a three-dimensional material, preparing a cross-sectional TEM sample right at the position of a filament would be experimentally challenging. On the other hand, there are other techniques that can more easily allow the conductive filament to be directly observed in such devices. We carried out conductive atomic force microscopy (c-AFM) measurement to image the conductive filament in SnOx. In this test, bias voltage is applied through the AFM tip to form the conductive filament in SnOx. c-AFM measurement then maps the current in the active region after the filament is set. Figure R1 shows the test structure (same structure as the actual device with the material layers in reverse order to allow c-AFM probing) used in the c-AFM measurement. The tip acts as a conductive probe to apply the voltage at the top of MoS2 layer, thereby setting the device into either the ON state or OFF state. A voltage of -5V is applied at the AFM tip (20 nm diameter) to set the device and the characteristic I-V curve shown in Figure R3 demonstrates the successful set process. After set, a small read voltage (-1V) is applied to read the current levels. In Figure R2, the green area indicates the region of higher current density (~105 pA) which is the location of the conductive filament. The filament diameter is about 20 nm. Besides, an obvious bubble-shape feature is also observed in the experimental process, when applying the voltage through the tip. This observation was also mentioned by other papers and could be explained by the oxygen accumulation in the filament region near the tip6,7, which supports the oxygen-vacancy type filament mechanism.

Figure R1 Test structure for c-AFM characterization.

Figure R2 c-AFM current mapping read at -1V after device is set. The green region indicates the location of the conductive filament region with higher current density.

Figure R3 **a**, First set curve (forming process). **b**, Subsequent set curve.

**Changes to supplementary information:** We have added Figure R1, R2 and R3 and the relevant discussions in Supplementary S3.

**Changes to manuscript:** We have revised the discussion about the device operation in the manuscript and added relevant references. These changes are highlighted in the revised manuscript.

**Comments 3:**

In addition to this, they just mentioned that both the conductivity of the MoS2 sheet itself and the interfacial barrier at MoS2/SnOx can be modulated by Vg, even without any energy band diagrams and references. And there is no optical/SEM image for top-view of the device. How can make the filament inside the SnOx when the voltage is applied to semiconducting (non-conducting) MoS2 and top metal electrode? The probing

ways and forming process are missing. This reviewer suggest that the authors must supplement the switching mechanism and the origin of the gate-tunable characteristics by including various experiments and discussions. Taken all together, the current manuscript is required to be significantly improved overall before it is published in Nature Communications. There are additional comments and suggestions that might help to improve the manuscript as below:

**Response:** We thank the reviewer for this comment. We included an optical image in Figure R4b for the top-view of a typical device below. As shown in this figure, the top electrode (TE) is deposited on top of  $\text{SnO}_x$ , and  $V_{\text{TE}}$  is applied to this electrode. The other metal electrode (BE) contact is  $\text{MoS}_2$  and is grounded. The gate bias  $V_g$  is applied to the Si back gate. Since  $\text{MoS}_2$  is a semiconducting material,  $V_g$  applied on the Si back gate can modulate its Fermi-level and charge density, such that its resistance can be modulated accordingly.

Figure R4 **a**, Schematic of  $\text{SnO}_x/\text{MoS}_2$  hetero-memristive device. **b**, Optical micrograph of a fabricated hetero-memristive device.

The device schematic and bias condition is shown in Figure R4a. The current path between the top electrode and the GND is a vertical flow through the filament in serial combination with the horizontal flow through the back-gated  $\text{MoS}_2$  layer. The resistance of the  $\text{MoS}_2$  layer can be modulated by the applied back-gate voltage, which in turn modulates the potential drop across the  $\text{SnO}_x$  filament switching layer. This potential drop across (and hence current flowing through) the  $\text{SnO}_x$  layer can still lead to filament formation and rupture in it. As the gate bias becomes more positive (or less positive/more negative), the  $\text{MoS}_2$  layer will be more conductive (or less conductive), which will in turn tune the effective potential drop across the  $\text{SnO}_x$  filament switching layer. A c-AFM mapping of conductive filament in an equivalent test structure is shown in Figure R2 in our response to the reviewer's previous comment.

The schematic band diagram along the current path, which is formed in the  $\text{MoS}_2$  layer in series with the filament in the  $\text{SnO}_x$  layer, is shown in Figure R5, where  $E_F=0$  is the Fermi level of GND,  $E_{F,\text{TE}}$  is the Fermi level of the top electrode, and  $F_{n,\text{MoS}_2}$  is the electron quasi-Fermi level in  $\text{MoS}_2$  at the position of the vertical filament contact. The short lines schematically represent the oxygen vacancy levels in the  $\text{SnO}_x$  layer. At low  $V_G$ , the semiconducting  $\text{MoS}_2$  layer has a low carrier density and thicker interface barrier, and as the gate voltage increases, the carrier density in  $\text{MoS}_2$  increases and the contact barrier thickness can be reduced by the gate modulation, which results in lower resistance. Hence, the gate bias modulates both the carrier density in the  $\text{MoS}_2$  layer as well as the properties of the  $\text{MoS}_2/\text{SnO}_x$  interfacial region, both of which in turn can influence the current and electric field inside the  $\text{SnO}_x$  layer in this device structure to affect the

filament dynamics. Similarly, other researchers have reported using voltage that is applied between semiconducting Si contact and top electrode to form filament inside SiO2 filament switching layer8.

Figure R5 Band diagram of SnOx/MoS2 hetero-memristive device under low and high VG.

In addition, following the reviewer's suggestion, we have included the data about the forming process below in Figure R6 (same figures as in Figure R3 above) obtained from the c-AFM measurement of equivalent test structure as shown in Figure R1. During the first switching cycle, a higher applied voltage is needed to form the initial filament as measured in the c-AFM characterization (shown in Figure R6 a). The forming voltage is around -6.5 V (the voltage is negative due to the reverse device configuration used in the c-AFM test (Figure R1)). After the initial forming process, the set voltage reduces to around -4.9 V for this test structure, lower than the forming voltage (Figure R6 b). The results confirm the forming process of the device.

Figure R6 **a**, first (forming) switching cycle. **b**, subsequent switching cycle.

**Changes to supplementary information:** We have added Figure R4b in Supplementary S1, Figure R5 in Supplementary S4, and Figure R6 in Supplementary S3 with relevant discussions.

#### Comments 4:

(1) The detailed probing methods and measurements in Fig. 1a should be clearly provided (e.g., an optical image). Also, the authors should identify electrical characteristics of the SnOx memristor without the insertion of MoS2 by applying Vg.

**Response:** We thank the reviewer for this comment. As discussed in our response to Comments 3 above, we have included an optical image in Figure R4b for the top-view of the device. As shown in this figure, the top electrode (TE) is deposited on top of SnOx, and VTE is applied to this electrode. The other metal electrode (BE) contact is MoS2 and is grounded. The gate bias Vg is applied to the Si back gate. Since MoS2 is a semiconducting material, Vg applied on the Si back gate can modulate its Fermi-level and charge density, such that its resistance can be modulated accordingly.

If the MoS2 under SnOx is completely removed, then the grounded electrode (contacted on the MoS2 sheet) will be disconnected, and the device would not have a conducting current path because now the SnOx layer would be connected with only a top electrode, an insulating SiO2 gate dielectric, and a Si gate.

If we do not insert MoS2 and replace it with a metal contact under SnOx, the device would lose its gate tunability because now Vg is completely screened by this metal film.

Moreover, if we do not consider using the gate, the intrinsic SnOx memristor with two metallic terminals has already been studied in the literature and numerous work has demonstrated its electrical characteristics including the unipolar memristive behavior4-6,9-12. These device, however, would not have the gate tunability.

#### Comments 5:

(2) In the sampling of exponential-class sigmoidal distribution, a statistical study was performed by switching from the high resistance state to the low resistance state for up to 2 second. This seem to be very slow. The authors should address the slow switching speed and discuss the effect of the slow sampling characteristics on the BM operation.

**Response:** We thank the reviewer for bringing up this point. Switching speed in our device mainly depends on the effective voltage that drops across the SnOx. Higher voltage can produce higher electric field and higher current, which can facilitate oxygen ion movement and speed up the filament dynamics and increase the switching speed.

Theoretically, the filament formation is controlled by microscopic ionic hopping and ionization process, whose rates can be expressed as,  $r = r_0 \exp(-\frac{E_b - d\epsilon}{k_B T})$ , where  $r_0$  is a rate constant,  $E_b$  is the barrier height,  $d$  is the effective hopping distance,  $\epsilon$  is the electric field along the hopping direction13. This equation shows that the speed of filament dynamics is approximately exponentially dependent on the electric field. This has been demonstrated both analytically and experimentally14,15.

As the applied voltage decreases, the electric field decreases, which results in exponential decrease of the hopping and ionization rates. In the sampling of exponential-class sigmoidal distribution that is demonstrated in this work, the applied voltage VTE is kept relatively low to allow for more robust longer term device operation, which leads to relatively slow switching in this work. To shorten the switching time, we could optimize the material and device sizes. Using thinner SnOx in this device would make the switching time shorter. Another approach is to select oxide material that may provide higher hopping and ionization rate than SnOx.

The sampling speed of the BM operation is mainly limited by the switching speed of the filament formation process in the SnOx/MoS2 structure. By optimization the device designs mentioned above, and together with scaling down the device size to reduce the parasitic, the sampling speed of the BM can be improved. In this work, we focus more on the demonstration of the new device concept, and further optimization of the device and BM speed performance will be our future work.

**Changes to supplementary information:** We have included the discussion on the switching speed in Supplementary S5.

#### Comments 6:

(3)  $V_g = -20\text{ V} - 30\text{ V}$  is very high, which can lead to the high power dissipation by  $V_g$  due to gate leakage current and gate charging process. Any comment about this issue?

**Response:** We thank the reviewer for this comment. In our device, the back gate has a 285 nm thermally grown SiO2 dielectric layer, which is also commonly used in the initial demonstration of many new two dimensional material based electronic device concepts16,17. The effective electric field introduced by  $V_g$  (in the range of -20 V to 30 V) is in fact quite low considering the thickness of the SiO2 dielectric layer. In our device, the gate leakage current is around a few pA as shown in Figure R7 (limited by noise level of the current measurement setup), which is very low if compared to the device currents. This low gate leakage current is not expected to lead to strong gate charging and high power dissipation.

The  $V_g$  can be scaled down by reducing the gate dielectric thickness and/or the use of high- $k$  dielectric. We would like to mention that the main focus of this work is on the initial demonstration of the new device concepts and its application, and it will be our future work to further optimize the performance metrics of the device.

Figure R7 Drain current and gate current of a hetero-memristive device during set process.

#### Comments 7:

(4) This reviewer did not find the endurance parameter in this manuscript. Because the multiple SET/RESET operation is required for the flip of Boolean value, the endurance seems to be important parameter. Any comment about this?

**Response:** Since this work focuses the initial demonstration of new device concept, we did not optimize the device in terms of the endurance metrics. On the other hand, the device endurance is still reasonable even without too much optimization. The endurance of the device switching is shown in Figure R8. Due to the limitations in our endurance characterization capabilities, the device is measured up to 5000 switching cycles. The data is also extrapolated to the  $10^6$  cycles mark. The set pulse is 5V and the read pulse is 1V. the reset pulse is -5 V and the read pulse is 1V.

Figure R8 Endurance measured up to 5000 consecutive switching cycles.

**Changes to supplementary information:** We have added Figure R8 in Supplementary S2.

**Comments 8:**

(5) Can the authors comment any advantage and benefit of the exponential-class sigmoidal for the BM applications?

**Response:** We thank the reviewer for this comment. The advantages and benefits of the exponential-class sigmoidal distribution for the BM applications could be summarized as following:

1. Sigmoid function is one of the most popular activation functions for the neural networks. The main reasons are (a) it is monotonic and continuous (b) it is differentiable and easy to compute the gradient which is always required for training.

2. The exponential-class sigmoidal sampling of the device fully captures the behavior of stochastic neuron, which is the basic composite of Boltzmann machine construction by definition. Its sampling distribution arising from the intrinsic randomness and energy distribution in the ionic motions can emulate the thermodynamic effect on neural behavior. The stochastic neurons may fire or not (flip the state or not) in response to the input signals and the firing rate is equivalent to the probability of successful SET occurrence following exponential-class sigmoidal distribution18,19.

3. Boltzmann machine is named after Boltzmann distribution, which is fundamental in many real-world physical phenomenon20-22. The exponential-class sigmoidal also resembles the Fermi-Dirac distribution that

determines the random ion movements in physical systems. Also, it is suitable for this framework to describe and solve the other problems that also follow the Fermi-Dirac distribution, which is general in real physics and device simulation23.

4. A concept of effective temperature is naturally introduced by the fundamental theory of BM implementation22. The device demonstrated here that is capable of generating exponential-class sigmoidal statistics provides the possibility of dynamically tuning the effective temperatures in the BM process. Thus, various “cooling” strategies, aka. simulated annealing method, could be implemented and optimized in the BM process for a better and faster convergence24.

#### **Comments 9:**

(6) The authors described the BM operation as a function of run or iteration number. How long the one run or iteration is taken? Because of the slow switching process of the fabricated device, the BM system can be further slow. The author should address this comment.

**Response:** We thank the reviewer for this comment. As has been addressed in our response to Comments 5, the switching speed of the BM is mainly limited by the SnOx/MoS2 device. In this work, each iteration of the BM operation takes around 3 seconds as the switching process of the fabricated device costs most of the running time in each iteration. In the MAX-SAT application we demonstrated in the manuscript, six such fabricated devices are required to form the DUTs, and they could be run in parallel simultaneously. In this MAX-SAT problem, we include six different Boolean variables and compose five clauses in total. As shown in Figure 3 (d-e), this optimization converges after ten iterations. This means the total runtime for solving such a representative MAX-SAT problem is around 30 seconds, and we also confirmed this in experimental tests.

As discussed in or response to Comments 5, by optimizing the material properties and device dimension, we can further improve the speed of the BM system.

#### **Comments 10:**

(7) The baseline obtained from the theoretical calculation or simulation need to be provided in the BM result to compare with the experimental results.

**Response:** We thank the reviewer for this very helpful comment. The simulation of the Boltzmann machine operation was conducted, and the simulated results are shown in Figure R9. Due to the stochastic nature of BM, we ran the simulation of BM for 50 times and plotted the averaged energy evolution under four “temperature cooling” strategies. The plot indicates the effect of temperature strategy on the optimization process of the BM, which matches well with the experimental results in Figure 4d.

Figure R9 The simulated result of energy evolution in BM under 4 different strategies for comparison to experimental result in Figure 4d.

**Changes to supplementary information:** We have added Figure R9 in Supplementary S10 with relevant discussion.

**Comments 11:**

(8) The authors insisted that an appropriate cooling strategy can lead to more efficient performance in the BM. Can the author suggest any guideline for the appropriate cooling strategy?

**Response:** We thank the reviewer for this comment. As discussed in the manuscript, implementing simulated annealing with an appropriate cooling strategy could lead to a better and faster convergence and prevent premature convergence in the BM.

A general guideline for the appropriate cooling strategy is to apply high effective “temperature” at the beginning of the optimization process to prevent premature convergence and applying lower effective “temperature” at the later stages of the optimization process to secure the convergence to global (or best local) optimum. The best detailed temperature cooling profile in an optimization process may vary based on the property of the optimization problem, such as the complexity of the problem, requirement for feasible convergence etc18,19. Despite it is more experiment-driven, a cooling strategy with linearly decreased (used in this paper) or a step-decreased temperature strategy (commonly used in annealing process) can be used as appropriate cooling strategy for initial try.

## Reviewer #2

### Comments 1:

This work reports simulated annealing and Boltzmann machine based on three-terminal hetero-memristors. The work are novel and will be of interest to others in the field. It can be published after addressing a few comments below:

**Response:** The authors would like to thank the reviewer for providing a careful review of our manuscript and for the positive assessment of this work. Below, we address the reviewer's comments point by point.

### Comments 2:

1, In the device structure, the authors used a thin SnSe oxidized  $\text{SnO}_x$  as the filament layer. How does the oxidation process affect the underlying  $\text{MoS}_2$ ? Will other oxide such as  $\text{HfO}_2$  function similarly?

**Response:** We thank the reviewer for this comment. The oxidation process does not affect the quality of the  $\text{MoS}_2$ . As discussed in the manuscript, a thin SnSe layer is first oxidized, and then the oxidized  $\text{SnO}_x$  is transferred to the top of  $\text{MoS}_2$  layer. The oxidization process would not affect the underlying  $\text{MoS}_2$ .

In such hetero-memristive device structure, using other oxide such as  $\text{HfO}_2$  should in principle function similarly. The filament formation process in  $\text{HfO}_2$  should also have similar dynamics and thus the device with  $\text{HfO}_2$  most likely can also provide sampling with exponential-class distributions. On the other hand, though the device concept in our work is applicable to other oxide, the bias conditions and the resulting parameters of the sampling distribution may change due to the different properties of different oxide materials.

### Comments 3:

2, Fig.2d, the  $T_{\text{eff}}$  approaches 50% when  $V_g = -20\text{V}$ , shall we expect the rapid increase when  $V_g$  goes more negative as the simulation shows? Is there any limit of the  $T_{\text{eff}}$ ?

**Response:** We thank the reviewer for this comment.  $T_{\text{eff}}$  characterizes the spread of the sigmoidal distribution, which depends on the resistance ratio between  $\text{SnO}_x$  and  $\text{MoS}_2$ . If  $V_g$  goes more negative than  $-20\text{V}$ , the resistance of  $\text{MoS}_2$  can increase because the Fermi level of carriers in  $\text{MoS}_2$  will approach the mid-gap level. As discussed in the manuscript, the spread of the sigmoidal distribution, in this case, would be wider due to a smaller portion of  $V_{\text{TE}}$  dropping across the  $\text{SnO}_x$  layer, which is represented by a higher  $T_{\text{eff}}$ .  $T_{\text{eff}}$  will keep increasing and eventually hit an upper-bound limit when the minimum conductance level of  $\text{MoS}_2$  is reached. Beyond this point, further increase in  $V_g$  would not generate higher  $T_{\text{eff}}$ .

In this work, we did not pursue the upper-bound limit of  $T_{\text{eff}}$ . It just needs to be high enough to introduce sufficient initial randomness in the simulated annealing process.

### Comments 4:

3, Fig.3, does it require each DUT to be exactly the same? Or not, will the randomness of DUT itself affect the machine implementation?

**Response:** We thank the reviewer for this comment. The DUTs do not need to be exactly the same. The BM in this work utilizes the stochastic properties from the DUTs and has high tolerance to the randomness

induced by variation of DUTs. The requirement for device uniformity is relatively relaxed in the BM as compared with other deterministic systems that are designed for solving combinatorial optimization problem, which is one significant advantage of the BM over the deterministic systems25,26.

#### Comments 5:

4, How does this approach compare with other methods using pure RRAM, MRAM or CMOS? What are the main merits?

**Response:** We thank the reviewer for this comment. Two-terminal RRAM and MRAM can be used for sampling the exponential-class distribution by changing different applied TE (top electrode)-BE (bottom electrode) voltages, but they do not have the capability to modulate the effective “temperature”  $T_{\text{eff}}$  dynamically and accurately. So, two-terminal RRAM and MRAM cannot introduce tunable  $T_{\text{eff}}$  in a single device that is a necessary property for implementing the simulated annealing process with controlled cooling strategies in the BM.

The CMOS implementation usually requires multiple device elements like transistors and capacitors to generate sigmoidal statistics. Many more devices and much more complicated circuits are required if we want to implement the BM and the associated simulated annealing process21,27. In comparison with the CMOS implementation, our approach only requires a single integrated device for the sampling functions.

#### Reference

- 1 Chang, W.-Y. *et al.* Unipolar resistive switching characteristics of ZnO thin films for nonvolatile memory applications. *Applied Physics Letters* **92**, 022110 (2008).
- 2 Goux, L. & Spiga, S. Unipolar Resistive-Switching Mechanisms. *Resistive Switching: From Fundamentals of Nanoionic Redox Processes to Memristive Device Applications*, 363-394 (2016).
- 3 Buh, G.-H., Hwang, I. & Park, B. H. Time-dependent electroforming in NiO resistive switching devices. *Applied Physics Letters* **95**, 142101 (2009).
- 4 Hsu, C.-C., Chuang, P.-Y. & Chen, Y.-T. Resistive switching characteristic of low-temperature top-electrode-free tin-oxide memristor. *IEEE Transactions on Electron Devices* **64**, 3951-3954 (2017).
- 5 Hota, M. K., Caraveo-Frescas, J. A., McLachlan, M. & Alshareef, H. N. Electroforming-free resistive switching memory effect in transparent p-type tin monoxide. *Applied Physics Letters* **104**, 152104 (2014).
- 6 Tian, H. *et al.* A hardware Markov chain algorithm realized in a single device for machine learning. *Nature communications* **9**, 1-11 (2018).
- 7 Yang, J. J. *et al.* The mechanism of electroforming of metal oxide memristive switches. *Nanotechnology* **20**, 215201 (2009).
- 8 Xia, Q. in *2016 18th Mediterranean Electrotechnical Conference (MELECON)*. 1-4 (IEEE).
- 9 Nagashima, K., Yanagida, T., Oka, K. & Kawai, T. Unipolar resistive switching characteristics of room temperature grown SnO2 thin films. *Applied Physics Letters* **94**, 242902 (2009).
- 10 Jin, J. *et al.* Effects of annealing conditions on resistive switching characteristics of SnOx thin films. *Journal of Alloys and Compounds* **673**, 54-59 (2016).

- 11 Guo, J. *et al.* Highly reliable low-voltage memristive switching and artificial synapse enabled by van der Waals integration. *Matter* **2**, 965-976 (2020).
- 12 Almeida, S., Aguirre, B., Marquez, N., McClure, J. & Zubia, D. Resistive switching of SnO2 thin films on glass substrates. *Integrated Ferroelectrics* **126**, 117-124 (2011).
- 13 Guy, J. *et al.* in *2014 IEEE International Electron Devices Meeting*. 6.5. 1-6.5. 4 (IEEE).
- 14 Ielmini, D. Modeling the universal set/reset characteristics of bipolar RRAM by field-and temperature-driven filament growth. *IEEE Transactions on Electron Devices* **58**, 4309-4317 (2011).
- 15 Zhang, H. *et al.* Gd-doping effect on performance of HfO2 based resistive switching memory devices using implantation approach. *Applied Physics Letters* **98**, 042105 (2011).
- 16 Sangwan, V. K. *et al.* Multi-terminal memtransistors from polycrystalline monolayer molybdenum disulfide. *Nature* **554**, 500-504 (2018).
- 17 Yan, X., Wang, H. & Sanchez Esqueda, I. Temperature-dependent transport in ultrathin black phosphorus field-effect transistors. *Nano letters* **19**, 482-487 (2018).
- 18 Ramachandran, P., Zoph, B. & Le, Q. V. Searching for activation functions. *arXiv preprint arXiv:1710.05941* (2017).
- 19 Fukushima, K. Visual feature extraction by a multilayered network of analog threshold elements. *IEEE Transactions on Systems Science and Cybernetics* **5**, 322-333 (1969).
- 20 Ackley, D. H., Hinton, G. E. & Sejnowski, T. J. A learning algorithm for Boltzmann machines. *Cognitive science* **9**, 147-169 (1985).
- 21 Bojnordi, M. N. & Ipek, E. in *2016 IEEE International Symposium on High Performance Computer Architecture (HPCA)*. 1-13 (IEEE).
- 22 Li, G. *et al.* Temperature based restricted Boltzmann machines. *Scientific reports* **6**, 1-12 (2016).
- 23 Fischer, A. & Igel, C. in *Iberoamerican congress on pattern recognition*. 14-36 (Springer).
- 24 Kirkpatrick, S., Gelatt, C. D. & Vecchi, M. P. Optimization by simulated annealing. *Science* **220**, 671-680 (1983).
- 25 Cai, F. *et al.* Power-efficient combinatorial optimization using intrinsic noise in memristor Hopfield neural networks. *Nature Electronics* **3**, 409-418 (2020).
- 26 Zhang, T. *et al.* Tolerance of intrinsic device variation in fuzzy restricted Boltzmann machine network based on memristive nano-synapses. *Nano Futures* **1**, 015003 (2017).
- 27 Yi, W., Park, J. & Kim, J.-J. GeCo: Classification restricted Boltzmann machine hardware for on-chip semisupervised learning and Bayesian inference. *IEEE transactions on neural networks and learning systems* **31**, 53-65 (2019).

Reviewers' Comments:

Reviewer #1:

Remarks to the Author:

It seems some of the parts in the manuscript is appropriately revised and improved according to the reviewer's comments in the first round of revision. However, the additional experiments and discussion for the switching mechanism are still unclear. The references [1-6] cited in the rebuttal letter seem to be a rather improper because the used metal-oxide, observed switching behaviors, and device structure were not the same; other metal-oxide materials (ZnO in Ref [1], NiO in Ref [2], NiO in Ref [3]), bipolar switching behaviors of SnOx-based memristors (Ref. [4, 5]), and other switching characteristics ( $V_{\text{set}} < V_{\text{reset}}$ ) in SnOx/SnSe/SnOx junction (Ref [6]). In addition, the additional experiments the author performed need further discussion and explanation for supporting the author's argument. For this reason, I think that this manuscript needs to be more revised with some supporting data and additional discussion.

My concerns are provided as below:

1) The authors added several published references to support their claim, where the device structure was fundamentally different. As the authors already know, the different material combinations and device structure can strongly affect the switching principle and characteristics of memristors.

2) In the AFM experiment, the authors performed the SET process by directly probing the AFM tip of 20-nm diameter (what kinds of metal tip was used?). This means, from the reviewer's perspective, that the tip acts as the 20-nm diameter top electrode practically. For the more correct c-AFM measurement, I think that it is necessary to perform the c-AFM measurements after etching the top electrode of the set- and reset-completed device. Hence, the device may not have a 20-nm conductive filament. Additionally, the current map in Fig. R2 needs to add the colorbar for current-level, and the current of green region is too low ~105 pA. Furthermore, the right I-V curve in Fig. R3 does not exhibit the reset in the reverse sweep (-8 V → 0 V) despite the high current level, which is different from the switching characteristics in Fig. 1e ( $V_{\text{reset}} = -3$  V in negative scan).

3) The device schematic of the device needs to be revised. From the optical image in Fig. R4, it is better to consider the MoS2 lateral dimension for clarification. Also, the ground symbol needs to be revised. The three dashes should be orientated horizontally, with the shortest one placed at the bottom rather than vertically.

4) I think that the authors should more substantiate the origin of gate-tunable characteristics. Various experiments and discussions should be required to study the origin of gate-tunable characteristics clearly. The suggested band diagram needs to be provided with some evidence and references, such as bandgap.

5) I wonder how the forming process of the device used in the manuscript occurs. Moreover, I don't agree that the authors did properly address the high gate voltage. The MOS capacitor in the gate region needs the power to charge up, and this power is proportional to  $\frac{1}{2}C \times V^2 \times f$ . In addition, the amplitude of the used  $V_G$  in Fig. R7 should be presented, and the current level of  $I_D$  in Fig. R7 is much lower up to  $\sim 2$ -3 order, compared with I-V curves in Fig. 1e, 1f, and R3.

6) In the endurance test, what was the used pulse width? Why were the positive set and negative reset pulses? The unipolar switching behavior of the device can be programmed by the same voltage polarity. I wonder whether the extrapolation used in the endurance test could be reasonable. How many pulses were applied in the BM application?

I encourage the authors to address some issues concerning the fabricated three-terminal memristor, because the similar simulated annealing applications with two-terminal memristor were already reported (Shin et al., *IEDM IEEE*, pp. 3.3.1–3.3.4. (2018); Yang et al., *Sci. Adv.* 6, eaba9901 (2020)).

Reviewer #2:

Remarks to the Author:

The authors have addressed all the issues and the manuscript can be accepted for publication in Nature Communications.

## Point-by-point Response to Reviewers' Comments for manuscript NCOMMS-21-01017A

We are grateful to the reviewers and editors for their time and very helpful comments. In response, we address the comments by the reviewers in detail. We always quote the reviewers' comments first.

---

### Reviewer #1

It seems some of the parts in the manuscript is appropriately revised and improved according to the reviewer's comments in the first round of revision. However, the additional experiments and discussion for the switching mechanism are still unclear. The references [1-6] cited in the rebuttal letter seem to be a rather improper because the used metal-oxide, observed switching behaviors, and device structure were not the same; other metal-oxide materials (ZnO in Ref [1], NiO in Ref [2], NiO in Ref [3]), bipolar switching behaviors of SnOx-based memristors (Ref. [4, 5]), and other switching characteristics ( $V_{\text{set}} < V_{\text{reset}}$ ) in SnOx/SnSe/SnOx junction (Ref [6]). In addition, the additional experiments the author performed need further discussion and explanation for supporting the author's argument. For this reason, I think that this manuscript needs to be more revised with some supporting data and additional discussion.

**Response:** We would like to thank the reviewer for acknowledging the improvements made in the previous round of review, and for the additional constructive comments. Here, we address the remaining questions raised by the reviewer.

### Comments 1:

The authors added several published references to support their claim, where the device structure was fundamentally different. As the authors already know, the different material combinations and device structure can strongly affect the switching principle and characteristics of memristors.

**Response:** We thank the reviewer for the comment. Regarding the references [1-6] mentioned by the reviewer in the comments above, we agree with the reviewer that there is no published paper that uses the exact same materials and device structure as our work and it is impossible for us to find such references to cite, which in turn shows the novelty of our work. Following the reviewer's suggestion, we have made the following changes to the references:

1. Ref. [1-6] in previous response letter are removed as the reviewer has suggested.
2. We added a new reference. Although the device structure in this new Ref is not exactly the same as ours, it used similar material SnOx for the oxide layer as our device and also demonstrated a unipolar resistive switching behavior and the existence of oxygen-vacancy filament, which carries supporting values for our work that unipolar switching in SnOx memristors are feasible.

Newly added reference:

Kazuki Nagashima, Takeshi Yanagida, Keisuke Oka, and Tomoji Kawai. Unipolar resistive switching characteristics of room temperature grown SnO2 thin films. *Applied Physics Letters* **94**, 24 (2009).

**Changes to Manuscript:** We removed the references [32-34] (same as Ref. [1-3] in previous response letter) and added a new Ref. 32 to the manuscript.

## Comments 2:

In the AFM experiment, the authors performed the SET process by directly probing the AFM tip of 20-nm diameter (what kinds of metal tip was used?). This means, from the reviewer's perspective, that the tip acts as the 20-nm diameter top electrode practically. For the more correct c-AFM measurement, I think that it is necessary to perform the c-AFM measurements after etching the top electrode of the set- and reset-completed device. Hence, the device may not have a 20-nm conductive filament. Additionally, the current map in Fig. R2 needs to add the colorbar for current-level, and the current of green region is too low  $\sim 105$  pA. Furthermore, the right I-V curve in Fig. R3 does not exhibit the reset in the reverse sweep ( $-8 \text{ V} \rightarrow 0 \text{ V}$ ) despite the high current level, which is different from the switching characteristics in Fig. 1e ( $V_{\text{reset}} = -3 \text{ V}$  in negative scan).

**Response:** We appreciate the reviewer for this comment. We used a standard AFM tip with 10 nm Cr fully-coated on the front side, which has the same metal configuration as the device in the manuscript.

First, we would like to mention that it is impossible to accurately etch the top-electrode of the set- and reset-completed device without impacting the formed filament inside. As the reviewer has mentioned in his other comments, the metal contacts on both the top and bottom electrodes assist in the formation and stabilization of the filaments. During the etching process, the etchant would be highly reactive with the oxygen vacancies on the surface of  $\text{SnO}_x$  once the metal is totally move. Moreover, even if the filament could survive the etching process, the nature of the device heterostructure in which the filament is stabilized would have become very different without the top contact.

In comparison, our current setup performs the measurement in-situ with the same device configuration as in other parts of our manuscript and is probably the best feasible configuration to be tested with our cAFM system. Moreover, this method has been extensively to support the switching mechanism of RRAM in other works in the literature (*Adv. Mater.* 20, 1154–1159 (2008), *Nanotechnol.* 21, 339803 (2009). *Nat. Commun.* 8, 15173 (2017).), etc.).

Regarding the second part of the reviewer's question about the current level of the cAFM current mapping, we in fact intentionally kept the current level very low (voltage bias at  $-1 \text{ V}$  for the original test device) during the c-AFM test to ensure the probing current would not disturb the existing filaments. Here, following the reviewer's request, we included a new measurement from another test device (as shown in Figure RR1) performed at a bias voltage of  $-2 \text{ V}$  in this new device, which shows higher filament current level above  $10 \text{ nA}$ . A color bar is also added for the current mapping image.

Regarding the reset mentioned by the reviewer, the current compliance used here (and also in the previous Fig. R3) in the cAFM measurements ( $\sim 30 \mu\text{A}$ ) is not high enough to observe the reset in the reverse sweep. As shown in Figure 1e of the manuscript, it usually needs larger current ( $\sim 250 \mu\text{A}$ ) to observe such reset in the device.

**Changes to supplementary information:** We deleted the original Figure S4 and S5 in Supplementary and updated with a new Figure S4 with relevant discussion in Supplementary S3.

Figure RR1, **a**, First set curve (represents forming process) and subsequent set curve. **b**, Current maps read by -2V after device is set in the c-AFM characterization. The green (and bright) area shows the position of higher current density.

**Comments 3:**

The device schematic of the device needs to be revised. From the optical image in Fig. R4, it is better to consider the MoS2 lateral dimension for clarification. Also, the ground symbol needs to be revised. The three dashes should be orientated horizontally, with the shortest one placed at the bottom rather than vertically.

**Response:** We thank the reviewer for pointing this out. The device schematic shown in manuscript Fig. 1a (same as the previous Fig. R4) has been updated as shown below.

Figure RR2, Schematic of tin oxide/MoS2 hetero-memristive device.

**Changes to Manuscript:** We have updated Fig. 1a in the manuscript.

#### Comments 4:

I think that the authors should more substantiate the origin of gate-tunable characteristics. Various experiments and discussions should be required to study the origin of gate-tunable characteristics clearly. The suggested band diagram needs to be provided with some evidence and references, such as bandgap

**Response:** We appreciate the reviewer for this comment. Here, we include some additional discussion and references of bandgap in band diagram as the reviewer has suggested.

This band diagram (Figure RR3 below) describes the current path between the top electrode and the GND, which consists of the vertical flow through the filament and the flow through the MoS2 layer.  $E_{F,TE}$  is the Fermi level of the top electrode, and  $F_{n,MoS_2}$  is the electron quasi-Fermi level in MoS2 at the position of the vertical filament contact. The short lines indicate the oxygen vacancy levels in the SnOx layer. The MoS2 material used in this experiment has multiple layers. Its quasi-partial bandgap is  $E_g \approx 1.29$  eV [Ref. RR1]. The bandgap of SnOx thin films is between 3.7 and 4.1 eV [Ref. RR2], which is significantly larger than the bandgap of MoS2. The gate tuning effect of the device filament can be illustrated with the different voltage drop across the filament part under different  $V_G$  as shown in Figure RR3.

At low  $V_G$ , the carrier density in the semiconducting MoS2 layer is relatively low and the interface barrier is relatively thick, hence the MoS2 layer has a high resistance and can be considered as at the OFF state. At OFF state, only a small fraction of  $V_{TE}$  is applied on the vertical filament part.

At high  $V_G$ , the carrier density in MoS2 increases and the contact barrier thickness decreases due to the gate modulation, hence the MoS2 layer has a low resistance and can be considered as at the ON state. At ON state, most of  $V_{TE}$  is applied on the vertical filament part.

Figure RR3, Band diagram of SnOx/MoS2 hetero-memristive device under low and high  $V_G$ .

The references for MoS2 bandgap and SnOx bandgap are referenced here and also added in supplementary material:

[RR1] Kobayashi, K., & Yamauchi, J. (1995). Electronic structure and scanning-tunneling-microscopy image of molybdenum dichalcogenide surfaces. *Physical Review B*, 51(23), 17085.

[RR2] Ali, F., Pham, N. D., Bradford, J., Khoshshirat, N., Ostrikov, K., Bell, J., ... & Tesfamichael, T. (2018). Tuning of oxygen vacancy in sputter-deposited SnOx films for enhancing the performance of perovskite solar cells. *ChemSusChem*, 11(18), 3096-3103.

**Changes to supplementary information:** We have updated Figure S5 (original Figure S6) and relevant discussion and references in Supplementary S4.

**Comments 5:**

I wonder how the forming process of the device used in the manuscript occurs. Moreover, I don't agree that the authors did properly address the high gate voltage. The MOS capacitor in the gate region needs the power to charge up, and this power is proportional to  $\frac{1}{2}C \times V^2 \times f$ . In addition, the amplitude of the used  $V_G$  in Fig. R7 should be presented, and the current level of  $I_D$  in Fig. R7 is much lower up to ~2-3 order, compared with I-V curves in Fig. 1e, 1f, and R3.

**Response:** We thank the reviewer for raising this point. The device forming occurs when the bias across the  $\text{SnO}_x$  layer reaches a critical level. Figure RR1 shows both the forming and subsequent switching curves as measured by cAFM in the test device structure.

The relatively high gate bias voltage needed is due to the thickness of  $\text{SiO}_2$  gate dielectric layer being around 285nm. The gate voltage can be scaled down by reducing the gate dielectric thickness for practical applications. We would like to mention that the main focus of this work is on the initial demonstration of the new device concepts and its application. The further optimization of device performance and device dimensional scaling will be of our future work.

We also agree with reviewer that the MOS capacitor in the gate region will need the power to charge up to the required gate bias voltage, proportional to  $\frac{1}{2}C \times V^2 \times f$ . When the gate dielectric of the device is thick like in our case, the capacitance will reduce linearly but the gate voltage required to operate the device will increase. So the overall power to charge the MOS capacitor will increase since it has a linear relation with respect to the capacitance and quadratic relation with respect to the voltage. On the other hand, for our proposed application in the BM demonstration, the gate bias is used to tune the "effective temperature" parameter in the cooling strategies, which does not need to change very fast. Hence, the frequency  $f$  does not need to be high in practical applications. Change in "effective temperature" only need to occur over many iteration cycle periods. In fact, changing "effect temperature" too fast would not benefit the optimization convergence. For example, in our simulated anneal with cooling demonstration, we kept gate voltage bias at the same value for the first three iterations and then changed the gate voltage bias to another value for the remaining seven iterations. The power to charge up the MOS capacitor is estimated to be ~13.6 pW, which is only a very small fraction of the overall power consumption of the computation. And again, as we mentioned earlier, this power consumption to charge up the MOS capacitor can be further reduced by scaling down the gate dielectric thickness.

The  $V_g$  used in the original Fig. R7 is zero. The current curve difference between Fig. R7 and I-V figures in manuscript is due to difference in device dimension, device-to-device variation, and also the current compliance being set to  $10^{-7}\text{A}$  in the original Fig. R7 (while current compliance is at  $10^{-5}\text{A}$  in the manuscript). To alleviate these discrepancies in the comparison, we include a new plot Fig. RR4, which is from a newly fabricated device with similar geometries as the device in the manuscript. The  $V_g$  used is -25V. The gate leakage current remains low.

Figure RR4, Drain current and gate current of a hetero-memristive device during set process.

#### Comments 6:

In the endurance test, what was the used pulse width? Why were the positive set and negative reset pulses? The unipolar switching behavior of the device can be programmed by the same voltage polarity. I wonder whether the extrapolation used in the endurance test could be reasonable. How many pulses were applied in the BM application?

**Response:** We thank the reviewer for this comment. The pulse width we used in the endurance test is 1ms. We agree with the reviewer that the voltage pulses with the same polarity can program the unipolar device. However, the voltage pulses with different polarity can also program the device successfully because of its unipolar property. This is also a reasonable and practical way to implement in the endurance test for such devices.

The number of applied pulses in the BM application varies under different test conditions (e.g., different initial states of the BM, simulated annealing strategies, etc.) in solving a MAX-SAT problem. The number of applied pulses on average is around one hundred.

#### Comments:

I encourage the authors to address some issues concerning the fabricated three-terminal memristor, because the similar simulated annealing applications with two-terminal memristor were already reported (Shin et al., IEDM IEEE, pp. 3.3.1–3.3.4. (2018); Yang et al., Sci. Adv. 6, eaba9901 (2020)).

**Response:** Here, we would like to clarify that the three-terminal memristors offers unique and fundamentally new functionalities as compared to the two-terminal devices mentioned by the reviewer. The BM demonstration and the simulated annealing with cooling strategy are impossible to realize with two-terminal devices:

1. The statistical distribution profile and parameters cannot be precisely controlled in two-terminal devices. For example, in the first paper (Shin et al., IEDM IEEE, pp. 3.3.1–3.3.4. (2018)), the probability of the device switching from HRS to LRS was enabled by applying a single set pulse with pulse width  $\Delta t$  and was implemented. The distribution function of the stochasticity is unknown and cannot be precisely tuned.

2. In the second paper (Yang et al., Sci. Adv. 6, eaba9901 (2020)), its SA was implemented by introducing decaying noise (chaos) into a chaotic Hopfield network. They achieved the performances that resembles SA (but not exactly SA) by using the transient chaos to make network more likely to converge toward the global minimum. With our three-terminal device, SA can be implemented in a more straightforward way. Our temperature modulation during different “cooling strategies” can be accurate and precise by tuning the gate voltage. The idea of changing “effective temperature” is also more general and intuitive with the three terminal device, as compared to chaos implementation.

3. Boltzmann machine, by its definition, requires exponential-class statistical characteristics in its stochastic element. Our tin oxide/MoS2 hetero-memristors are capable of sampling exponential-class sigmoidal distributions with tunable parameters, which is very versatile activation functions for neural networks that can harvest the benefits of BM. The stochastic or chaotic characteristics in these two papers mentioned by the reviewer were not proved to exhibit exponential class features. They may be used to realize Hopfield network, but not specifically the Boltzmann machines.

On the other hand, the two papers based on two-terminal devices as mentioned by the reviewer are very good work on relevant topics. We included both of them as the references in our manuscript.

**Changes to Manuscript: We have added these two references as Ref. 46 and Ref. 47 to the manuscript.**

Reviewers' Comments:

Reviewer #1:

None